# Bicuspid Aortic Valve Alters Aortic Protein Expression Profile in Neonatal Coarctation Patients

**DOI:** 10.3390/jcm8040517

**Published:** 2019-04-16

**Authors:** Katie L. Skeffington, Andrew R. Bond, Safa Abdul-Ghani, Dominga Iacobazzi, Sok-Leng Kang, Kate J. Heesom, Marieangela C. Wilson, Mohamed Ghorbel, Serban Stoica, Robin Martin, M. Saadeh Suleiman, Massimo Caputo

**Affiliations:** 1Bristol Heart Institute, Research Floor Level 7, Bristol Royal Infirmary, Upper Maudlin Street, Bristol BS2 8HW, UK; katie.skeffington@bristol.ac.uk (K.L.S.); Andrew.Bond@bristol.ac.uk (A.R.B.); salghani@staff.alquds.edu (S.A.-G.); mdxdi@bristol.ac.uk (D.I.); M.Ghorbel@bristol.ac.uk (M.G.); M.Caputo@bristol.ac.uk (M.C.); 2Department of Congenital Heart Disease, King David Building, Upper Maudlin Street, Bristol BS2 8JB, UK; soklengkang@doctors.org.uk (S.-L.K.); mdzscs@bristol.ac.uk (S.S.); Rob.Martin@UHBristol.nhs.uk (R.M.); 3Proteomics Facility, University of Bristol, Bristol BS8 1TD, UK; K.Heesom@bristol.ac.uk (K.J.H.); Maz.Wilson@bristol.ac.uk (M.C.W.)

**Keywords:** aortic coarctation, bicuspid aortic valve, congenital heart disease

## Abstract

Coarctation of the aorta is a form of left ventricular outflow tract obstruction in paediatric patients that can be presented with either bicuspid (BAV) or normal tricuspid (TAV) aortic valve. The congenital BAV is associated with hemodynamic changes and can therefore trigger different molecular remodelling in the coarctation area. This study investigated the proteomic and phosphoproteomic changes associated with BAV for the first time in neonatal coarctation patients. Aortic tissue was collected just proximal to the coarctation site from 23 neonates (BAV; *n* = 10, TAV; *n* = 13) that were matched for age (age range 4–22 days). Tissue from half of the patients was frozen and used for proteomic and phosphoproteomic analysis whilst the remaining tissue was formalin fixed and used for analysis of elastin content using Elastic Van-Gieson (EVG) staining. A total of 1796 protein and 75 phosphoprotein accession numbers were detected, of which 34 proteins and one phosphoprotein (SSH3) were differentially expressed in BAV patients compared to TAV patients. Ingenuity Pathway Analysis identified the formation of elastin fibres as a significantly enriched function (*p* = 1.12 × 10^−4^) due to the upregulation of EMILIN-1 and the downregulation of TNXB. Analysis of paraffin sections stained with EVG demonstrated increased elastin content in BAV patients. The proteomic/phosphoproteomic analysis also suggested changes in inositol signalling pathways and reduced expression of the antioxidant SOD3. This work demonstrates for the first time that coarcted aortic tissue in neonatal BAV patients has an altered proteome/phosphoproteome consistent with observed structural vascular changes when compared to TAV patients.

## 1. Introduction

Coarctation of the aorta (CoA) is a common congenital defect whose overall incidence is 5–8% of all congenital cardiac anomalies [1]. The majority of CoA patients also have a bicuspid aortic valve (BAV). CoA is a known risk factor for aortic complications including aneurysm formation or aortic dissection [2,3], and the presence of BAV further increases the risks for CoA patients. Studies have demonstrated that CoA patients with BAV have increased risk of aortic aneurysm [4], suboptimal improvement in left ventricular function following repair [5] and greater long-term cardiovascular morbidity [6]. The coexistence of CoA and BAV has been demonstrated to significantly affect aortic blood flow haemodynamics, for example increasing turbulent flow and affecting indices of shear stress both upstream and downstream of the coarctation [7]; such haemodynamic changes may at least partly explain the heightened cardiovascular risks in these patients. It is suggested that surgical correction of CoA repairs the anatomical narrowing but does not address the associated vasculo- and valvulopathy [8].

BAV can present in different phenotypes which may be related to congenital, genetic, and/or connective tissue abnormalities [9]. Discrimination between BAV subtypes may potentially provide clinical and prognostic information in patients with coarctation of the aorta [10]. Interestingly, induced loss-of-function in murine *Nos3* (which encodes endothelial nitric oxide synthase type 3) or in human *GATA5* (which encodes a Zn-finger transcription factor) have both been reported to associate with BAV phenotype [11,12]. Moreover, *NOTCH1* mutations are associated with cardiovascular malformations, and many of the patients with CoA tend to have mutations in the *NOTCH1* gene [13].

However, a comparison of the whole proteome of CoA patients with and without BAV has not been performed. Studies using tissue from patients undergoing aortic aneurism surgery have demonstrated significant differences in the proteome of BAV patients compared to TAV patients (e.g., [14]), however a similar analysis has not been applied to neonatal CoA patients. The effects of factors (such as altered blood flow haemodynamics) which may affect the aortic proteome and phosphoproteome are likely to still be evolving in the weeks after birth. In this study we therefore compared the proteome and phosphoproteome of aortic tissue from very young (less than three weeks old) CoA patients with and without BAV, thus providing a unique insight into the vascular molecular remodelling occurring in neonatal CoA patients as a result of congenital valve malformation.

## 2. Experimental Section

### 2.1. Patients and Tissue Collection

Tissue was collected just proximal to the coarctation site from neonatal patients undergoing congenital surgery including repair of the aortic coarctation. The study was conducted in accordance with the declaration of Helsinki, and the protocol was approved by the North Somerset and South Bristol Research Ethics Committee (REC 07/H0106/172). Full informed consent was obtained from parents prior to admission for operation. Aortic tissue from the coarctation area of half the patients was snap frozen in liquid nitrogen before being stored at −80 °C (TAV: *n* = 5, aged 10 ± 2 days (mean ± SEM). BAV: *n* = 7, aged 10 ± 2 days). Aortic tissue from the remainder of the patients was fixed in 10% formalin before being transferred to PBS for storage (TAV: *n* = 5, aged 7 ± 1 days. BAV: *n* = 6, aged 9 ± 2 days).

### 2.2. Sample Preparation

Proteins were extracted in radio-immuno-precipitation assay (RIPA) buffer (1% NP-40, 0.5% sodium deoxycholate, 0.1% SDS, in PBS) containing phosphatase and protease inhibitors, and quantified using the Bradford assay. Aliquots of 100 µg of 10 samples per experiment were digested with trypsin (2.5 µg trypsin per 100 µg protein; 37 °C, overnight), labelled with Tandem Mass Tag (TMT) 10Plex reagents according to the manufacturer’s protocol (Thermo Fisher Scientific, Loughborough, LE11 5RG, UK), and the labelled samples pooled.

For the total proteome analysis, aliquots of 50 µg of the pooled sample were evaporated to dryness and re-suspended in buffer A (20 mM ammonium hydroxide, pH 10) prior to fractionation by high pH reversed-phase chromatography using an UltiMate 3000 liquid chromatography system (Thermo Fisher Scientific). The sample was loaded onto an XBridge BEH C18 Column (130 Å, 3.5 µm, 2.1 × 150 mm, Waters, UK) in buffer A and peptides eluted with an increasing gradient of buffer B (20 mM Ammonium Hydroxide in acetonitrile, pH 10) from 0–95% over 60 min. The resulting fractions were evaporated to dryness and re-suspended in 1% formic acid prior to analysis by nano-LC MSMS using an Orbitrap Fusion Tribrid mass spectrometer (Thermo Fisher Scientific).

For the phosphoproteome analysis, the remainder of the TMT-labelled pooled sample was evaporated to dryness and subjected to TiO_2_-based phosphopeptide enrichment according to the manufacturer’s instructions (Pierce). The phospho-enriched sample was evaporated to dryness and then re-suspended in 1% formic acid prior to analysis by nano-LC MSMS using an Orbitrap Fusion Tribrid mass spectrometer (Thermo Fisher Scientific).

### 2.3. Nano-LC Mass Spectrometry

High pH RP fractions (total proteome analysis) or the phospho-enriched fraction (phospho-proteome analysis) were further fractionated using an Ultimate 3000 nano-HPLC system in line with an Orbitrap Fusion Tribrid mass spectrometer (Thermo Scientific). In brief, peptides in 1% (vol./vol.) formic acid were injected onto an Acclaim PepMap C18 nano-trap column (Thermo Scientific). After washing with 0.5% (vol./vol.) acetonitrile 0.1% (vol./vol.) formic acid, peptides were resolved on a 250 mm × 75 μm Acclaim PepMap C18 reverse phase analytical column (Thermo Scientific) over a 150 min organic gradient, using seven gradient segments (1–6% solvent B over 1 min, 6–15% B over 58 min, 15–32% B over 58 min, 32–40% B over 5 min, 40–90% B over 1 min, held at 90% B for 6 min and then reduced to 1% B over 1 min) with a flow rate of 300 nL·min^−1^. Solvent A was 0.1% formic acid and Solvent B was aqueous 80% acetonitrile in 0.1% formic acid. Peptides were ionized by nano-electrospray ionization at 2.0 kV using a stainless-steel emitter with an internal diameter of 30 μm (Thermo Scientific) and a capillary temperature of 275 °C.

All spectra were acquired using an Orbitrap Fusion Tribrid mass spectrometer controlled by Xcalibur 2.0 software (Thermo Scientific) and operated in data-dependent acquisition mode using an SPS-MS3 workflow. FTMS1 spectra were collected at a resolution of 120,000, with an automatic gain control (AGC) target of 200,000 and a maximum injection time of 50 ms. The Top N most intense ions were selected for MS/MS. Precursors were filtered according to charge state (to include charge states 2–7) and with mono-isotopic precursor selection. Previously interrogated precursors were excluded using a dynamic window (40 s +/− 10 ppm). The MS2 precursors were isolated with a quadrupole mass filter set to a width of 1.2 m/z. ITMS2 spectra were collected with an AGC target of 5000, max injection time of 120 ms and CID collision energy of 35%.

For FTMS3 analysis, the Orbitrap was operated at 60,000 resolution with an AGC target of 50,000 and a max injection time of 120 ms. Precursors were fragmented by high-energy collision dissociation (HCD) at normalised collision energy of 55% to ensure maximal TMT reporter ion yield. Synchronous Precursor Selection (SPS) was enabled to include up to five MS2 fragment ions in the FTMS3 scan.

### 2.4. Data Processing and Analysis

The raw data files were processed and quantified using Proteome Discoverer software version 1.4 (Thermo Scientific) and peptide sequences searched against the Uniprot Human database (134,169 sequences) using the SEQUEST algorithm. Peptide precursor mass tolerance was set at 10 ppm, and MS/MS tolerance was set at 0.6 Da. Search criteria included oxidation of methionine (+15.9949) as a variable modification and carbamido-methylation of cysteine (+57.0214) and the addition of the TMT mass tag (+229.163) to peptide N-termini and lysine as fixed modifications. For the phosphoproteome analysis, phosphorylation of serine, threonine and tyrosine (+79.966) were also included as variable modifications. Searches were performed with full tryptic digestion and a maximum of one missed cleavage was allowed. The reverse database search option was enabled and all peptide data was filtered to satisfy false discovery rate (FDR) of 5%.

Any proteins or phosphoproteins with more than one missing value per sample group were excluded from the analysis [15]. Putative uncharacterized proteins or proteins with either accession numbers representing cDNA with weak similarity were also excluded. For the proteomics data, values for each of the proteins identified are presented as a ratio to the internal standard (a pool of all samples) and represent the median of the measured peptide(s) for each protein. Fold changes between BAV and TAV patients were calculated, and high-fold differences (fold decrease/increase greater than ±1.3 identified. The value of 1.3 was used as a cut off as it has previously been used in other, similar studies, for example [16]. Significant changes were determined by an unpaired Student’s *t*-test (*p* < 0.05 being significant). Volcano plots were created by plotting log_2_ (fold difference) on the horizontal axis and −log_10_ (*p*-value) on the vertical axis.

Gene ontology (GO) enrichment analysis of differentially expressed proteins was performed using QuickGO software. Differentially expressed proteins and phosphoproteins were inputted into Ingenuity Pathway Analysis software (IPA version 46901286, Qiagen, Aarhus, Denmark) to determine significantly enriched canonical pathways, diseases and functions (*p*-value of overlap calculated by Fisher’s exact test right tailed).

### 2.5. Elastin Staining Analysis

Fixed aortic tissue was embedded in paraffin and sectioned at 5 µm. Three to five sections from each patient were stained with Elastic Van-Gieson stain (EVG). Elastin density was measured at 40× magnification on four randomly selected areas of each section using ImageJ software (National Institute of Health, Bethesda, MD, USA, Version 1.46). The results were analysed with an unpaired Student’s *t*-test, significance was accepted when *p* < 0.05.

## 3. Results

A total of 1796 protein accession numbers were detected, of which 34 were significantly differentially expressed between BAV and TAV patients (Figure 1A, Table 1). Of these, two proteins (TNC and EMILIN1) were significantly up-regulated in BAV patients compared to TAV patients, whilst the remainder were significantly down regulated.

The differentially expressed proteins were functionally classified by Gene Ontology analysis under the three main categories of GO analysis (i.e., biological process, cellular component and molecular function). Protein binding and superoxide dismutase activity were the most represented GO terms within molecular functions, whilst cytoplasm and protein stabilization were the most represented cellular component and biological process respectively (Figure 2). Superoxide radical degeneration was also identified by IPA Ingenuity pathway analysis as one of two significantly (*p* < 0.05) enriched canonical pathways, the other being D-myo-inositol (1,4,5) trisphosphate biosynthesis (Table 2).

Table 3 shows the top five most significantly enriched diseases or functions overall, as well as all significantly enriched diseases and functions falling under the IPA category ‘Cardiovascular system Development and Function’. Aortic valve morphogenesis was not surprisingly identified as being highly significant (*p* = 3.44 × 10^−4^). Elastin fibre formation was the second most significantly enriched function overall (*p* = 1.12 × 10^−4^). The two proteins associated with this change were TNXB and EMILIN-1, the latter being one of only two proteins upregulated in BAV patients compared to TAV patients. Histological measurements in separate cohorts of patients confirmed a significant difference in elastin content between the two groups, with greater elastin content in the aortas of BAV patients (Figure 3).

A total of 75 phosphorylated proteins were identified, some with multiple phosphorylation sites, resulting in 90 phosphorylation site matches. Only one, protein phosphatase slingshot homolog 3, was differentially expressed between BAV patients and TAV patients (Table 4), resulting in significant enrichment in a number of pathways involved in inositol metabolism (Table 5).

## 4. Discussion

To the best of our knowledge, this is the first comparison of the neonatal aortic proteome of CoA patients with BAV compared to CoA patients with a normal valve. Although the anatomy of the coarcted aorta appears the same in the two groups of patients, this study provides evidence for proteomic and histological differences associated with the presence of an abnormal valve.

### 4.1. Proteomic and Structural Analysis Identify Changes in Elastin as the Main Structural Aortic Alteration Associated with BAV

Prostaglandin E1 is often administered to neonatal CoA patients to maintain the patency of the ductus arteriosis, hence reducing the total velocity of blood flow through the aortic valve [17]. However, it has been suggested that the coexistence of CoA and BAV increases turbulent flow and shear stress both upstream and downstream of the coarctation area compared to patients with CoA alone [7]. Such changes in blood haemodynamics may, in turn, cause aortic molecular and structural remodelling. Two of the proteins which demonstrated significantly altered expression in this study were EMILIN-1, whose expression was upregulated in BAV patients compared to TAV patients, and TENASCIN-X, whose expression was downregulated in BAV patients. Canonical pathway analysis suggested that these changes indicate significant differences in elastin fibre formation between the two groups, and histological analysis in a separate subset of patients did indeed demonstrate significantly increased elastin content in BAV patients compared to TAV patients.

EMILIN-1 is a glycoprotein which associates with elastic fibres at the interface of elastin and microfibrils. It is known to be involved in the creation of molecular interactions between elastic fibre components and therefore to play an important role in elastogenesis [18]. This is confirmed by the finding that EMILIN-1 deficiency causes elastin fragmentation [18,19], thus the observed increase in expression of EMILIN-1 in BAV patients in this study is consistent with greater elastin density.

TENASCIN-X is an extracellular matrix protein whose expression was downregulated in BAV patients. A deficiency in TENASCIN-X protein has been reported to be one cause of Ehlers-Danlos syndrome (EDS), and patients with this form of EDS can suffer from abnormal dermal elastin fibre morphology which is suggested to be caused by altered elastin assembly and stability rather than altered degradation of elastin [20]. The fact that elastin fibre abnormalities are not present in other forms of EDS has been purported as evidence that the elastin fibre deficiencies are a direct result of the deficiency of the TENASCIN-X protein [21].

Interestingly, another protein from the TENASCIN family, TENASCIN-C was also differentially expressed between the two patient groups and was the only other protein besides EMILIN-1 to be upregulated in BAV patients compared to TAV patients. Previous studies have also found concurrent increases in TENASCIN-C and elastin, for example in patients with lung fibrosis [22]. However, other studies have found conflicting results, for example one study demonstrated that increases in TENASCIN-C are associated with increased elastin calcification and degradation [23]. The precise nature of any causative relationships between the TENASCIN-C and elastin is therefore unclear and may differ between organs.

Elastin is an important structural component of the aorta, creating elasticity in the walls of the vessel and therefore allowing buffering of the pressure changes generated by the cardiac cycle. Experiments using cultured human smooth muscle cells have demonstrated that elastin synthesis can be affected by changes in strain or application of pulsatile flow [24,25]. A study in adult BAV patients demonstrated greater elastin degradation in aortic areas with increased wall shear stress compared to adjacent areas with normal wall shear stress in the same patient [26]. Thus, it is possible that haemodynamic changes resulting from the presence of BAV are responsible for the correlation between elastin abnormalities and the presence of BAV observed in the current study.

### 4.2. Inositol and Oxidative Stress Signalling Pathways are Altered in BAV Patients

The enzyme superoxide dismutase 3 (SOD3) was significantly under-expressed in the aortas of BAV patients compared to TAV patients. In adult humans, the BAV phenotype has been shown to associate with increased levels of aortic oxidative stress [27]. One explanation suggested for this is that mechanical stretch of blood vessels is known to result in increased superoxide production via various pathways including activation of NADPH oxidase, activation of xanthine oxidase and an increase in eNOS signalling [28]. Interestingly, despite the presence of increased oxidative stress in aortas of adults with the BAV phenotype, the tissues fail to show increased superoxide dismutase activity—total SOD activity was demonstrated to be similar between BAV and TAV patients [27], whilst multiple studies have demonstrated SOD3 expression to be reduced in adult BAV patients compared to TAV patients [29,30]. Data from the current study indicates, to the best of our knowledge for the first time, that SOD3 expression is also associated with BAV in neonatal CoA patients.

Changes suggestive of altered inositol signalling correlating with the presence of BAV were also found in both the proteomic and phosphoproteomic analyses. Inositide molecules are involved in many aspects of cellular signalling within the cardiovascular system and have been linked to modulation of aortic structure and aortic aneurysm formation [31]. In the current study, Phospholipase C (PLC) was significantly downregulated in BAV patients, whilst altered phosphorylation of SSH3 affects many inositol signalling pathways. Few studies have investigated changes in inositol signalling associated with BAV, although changes in the abundance of different isoforms of protein kinase C (PKC), which interacts with inositol signalling molecules, have been described in adult BAV patients [32].

Limitations of the study include the fact that we do not have aortic tissue from healthy neonates, clearly this would be the ideal control group but is very difficult if not impossible to obtain. It would also have been ideal to carry out both the proteomic and histological analyses in every patient, however unfortunately the size of tissue samples which can be obtained from neonatal patients is extremely small, making it very difficult to perform both analyses for every patient. Slight regional differences in the exact location of the tissue collection may unfortunately have introduced variation into our analyses; regional differences in molecular remodelling in CoA patients could be a focus of future studies. Additionally, it would have been beneficial to validate the changes in protein expression using Western blotting.

## 5. Conclusions

Neonatal CoA patients with a bicuspid aortic valve show less improvement in ventricular function following repair and greater long-term cardiovascular risks compared to CoA patients with a normal, tricuspid aortic valve [4,5,6]. This study found significant differences in the aortic proteome of neonatal CoA patients dependent on the morphology of the aortic valve. Compared to patients with a normal valve, BAV patients demonstrated increased elastin content, changes in the expression of proteins involved in response to oxidative stress, and alterations in proteins involved in inositol signalling pathways. Whilst previous studies which have compared BAV and TAV patients have all taken place in adults, this study is the first to investigate the effects of a bicuspid aortic valve in neonatal children with CoA. Improved understanding of molecular and structural remodelling of the aorta in neonatal patients with BAV will help in the design of long-term therapeutic targets.

## Figures and Tables

**Figure 1 jcm-08-00517-f001:**
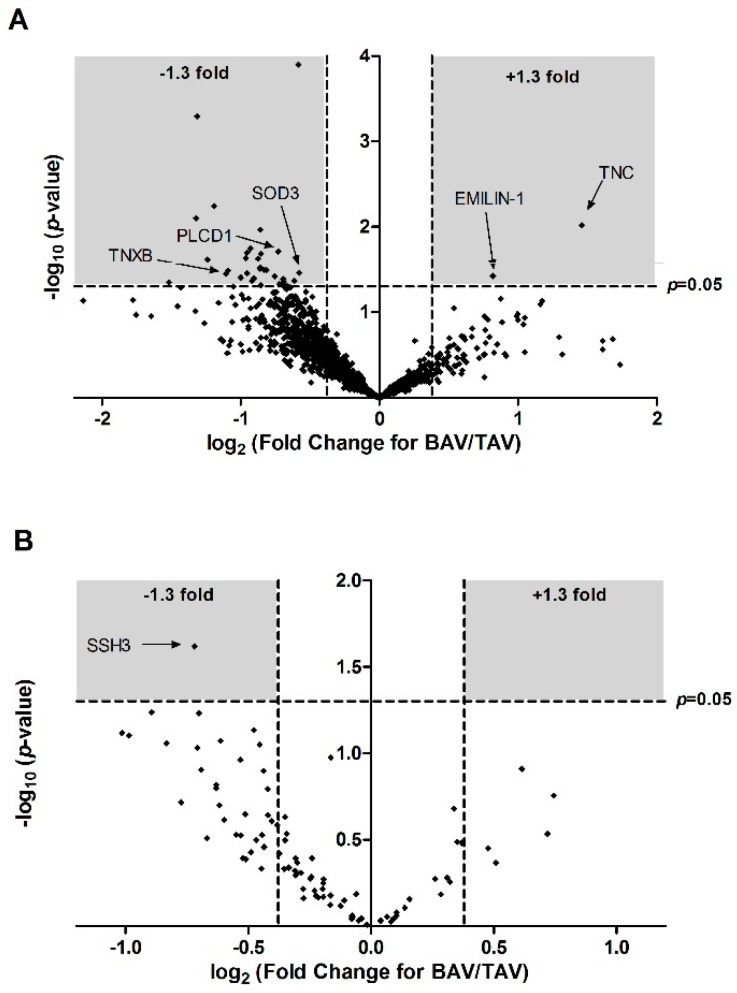
Volcano plot of proteins (**A**) and phosphoproteins (**B**) quantified in neonatal patients with bicuspid (BAV) verses patients with normal tricuspid (TAV). Each point represents the log_2_ (fold change) between the groups plotted against the associated significance of this change. Proteins significantly altered (±1.3 fold change, *p* < 0.05) are shown in the grey shaded areas. TNXB: Tenascin-X; PLCD1: Phosphoinositide phospholipase C; SOD3: Extracellular superoxide dismutase 3; EMILIN-1: elastin microfibril interfacer 1; TNC: Tenascin C; SSH3: Protein phosphatase slingshot homolog 3; BAV: bicuspid aortic valve; TAV: tricuspid aortic valve.

**Figure 2 jcm-08-00517-f002:**
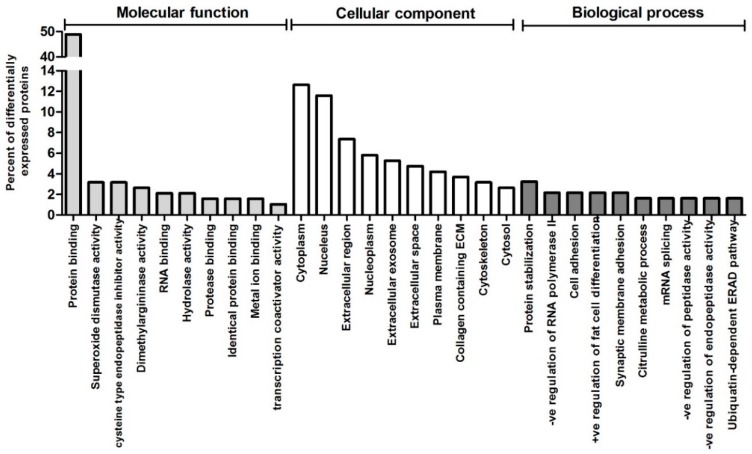
Functional categorization by Gene Ontology (GO) analysis of the proteins differentially expressed between BAV and TAV patients. The top 10 GO terms in each of the three main categories of GO classification (molecular function, cellular component and biological process) are displayed. The y axis represents the percentage of a specific category of proteins within the main category.

**Figure 3 jcm-08-00517-f003:**
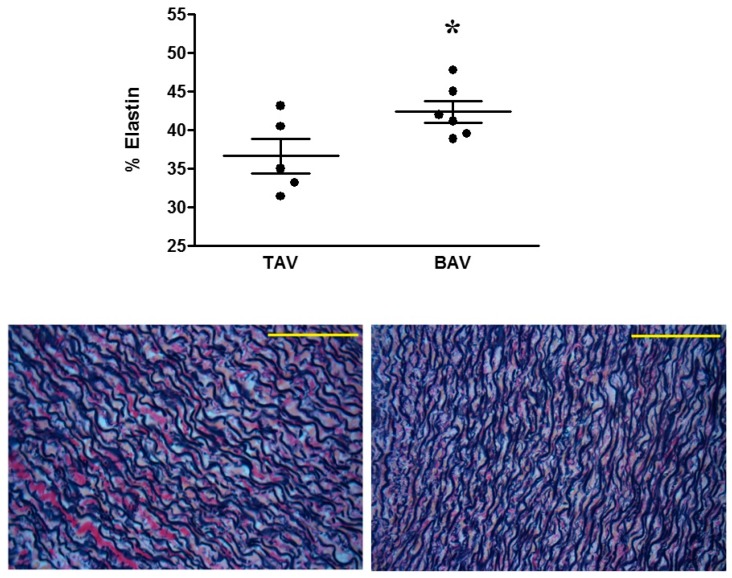
Elastin content in TAV vs. BAV patients. * *p* < 0.05, Student’s *t*-test. Representative elastin staining from TAV (**left**) and BAV (**right**) patients. Scale bars represent 40 µm.

**Table 1 jcm-08-00517-t001:** Proteins differentially (fold decrease/increase > ±1.3)) and significantly (*p* < 0.05) expressed in coarcted aorta from neonatal BAV patients compared to neonatal TAV patients.

Accession #	Gene ID	Description	Mean		SEM		Fold Change (BAV/TAV)	*p*-Value	log_2_ (Fold Change)	−log_10_ (*p*-Value)
			BAV	TAV	BAV	TAV				
F5H7R9	PTMS	Parathymosin (Fragment)	0.52	1.49	0.13	0.47	0.35	0.045	−1.52	1.35
A0A024R1R8	hCG_2014768	HCG2014768, isoform CRA_a	1.79	4.47	0.4	0.79	0.40	0.008	−1.32	2.10
Q15847	ADIRF	Adipogenesis regulatory factor	0.50	1.24	0.09	0.13	0.40	0.001	−1.31	3.29
A0A087WY58	NEK3	Serine/threonine-protein kinase Nek3	4.09	9.67	1.67	0.79	0.42	0.024	−1.24	1.61
P23468	PTPRD	Receptor-type tyrosine-protein phosphatase delta	2.63	6.00	0.59	0.78	0.44	0.006	−1.19	2.24
A0A087WWA5	TNXB	Tenascin-X	0.73	1.58	0.11	0.39	0.46	0.036	−1.11	1.45
A0A087WYN4	TTC25	Tetratricopeptide repeat protein 25	0.49	1.05	0.11	0.22	0.47	0.033	−1.09	1.48
B4DVR4	-	cDNA, FLJ60912, highly similar to Vinexin	1.91	3.81	0.35	0.82	0.50	0.039	−1.00	1.41
Q96CN7	ISOC1	Isochorismatase domain-containing protein 1	1.26	2.45	0.29	0.34	0.51	0.023	−0.96	1.63
P51610	HCFC1	Host cell factor 1	0.99	1.91	0.17	0.32	0.52	0.020	−0.95	1.70
O76070	SNCG	Gamma-synuclein	0.67	1.30	0.18	0.18	0.52	0.035	−0.95	1.46
E9PNK6	TPD52L1	Tumour protein D53	1.43	2.72	0.29	0.35	0.53	0.018	−0.93	1.75
B7Z650	-	cDNA, FLJ58685, highly similar to Homo sapiens echinoderm microtubule associated protein like 1	1.15	2.17	0.25	0.40	0.53	0.043	−0.92	1.37
H7C4C5	MAP4	Microtubule-associated protein (Fragment)	0.87	1.63	0.20	0.26	0.53	0.040	−0.90	1.40
O76024	WFS1	Wolframin	1.29	2.36	0.25	0.33	0.54	0.024	−0.88	1.62
Q4ZG81	FLJ20701	Putative uncharacterized protein FLJ20701 (Fragment)	0.92	1.66	0.22	0.16	0.55	0.031	−0.86	1.51
Q9Y4G6	TLN2	Talin-2	1.04	1.88	0.20	0.28	0.55	0.030	−0.86	1.52
A0A024RAR8	ARTS-1	Type 1 tumour necrosis factor receptor shedding aminopeptidase regulator, isoform CRA-a	0.87	1.58	0.14	0.18	0.55	0.011	−0.86	1.97
Q59GL1	-	Synaptotagmin binding, cytoplasmic RNA interacting protein variant (Fragment)	2.65	4.8	0.59	0.78	0.55	0.048	−0.86	1.31
A8K8F9	PLCD1	Phosphoinositide phospholipase C	0.81	1.47	0.15	0.18	0.55	0.021	−0.85	1.69
B1AJY5	PSMD10	26S proteasome non-ATPase regulatory subunit 10	1.62	2.88	0.35	0.35	0.56	0.032	−0.83	1.50
BD4L66	-	cDNA FLJ56297, highly similar to Rattus norvegicus ubiquitin-conjugating enzyme E2Z (PUTATIVE) (Ube2z)	2.04	3.60	0.36	0.54	0.57	0.032	−0.82	1.49
F2Z2V0	CPNE1	Copine-1 (Fragment)	0.86	1.45	0.11	0.25	0.59	0.038	−0.75	1.42
Q9H7C9	AAMDC	Mth938 domain-containing protein	1.39	2.31	0.25	0.16	0.60	0.020	−0.73	1.71
O95865	DDAH2	N(G), N(G)-dimethylarginine dimethylaminohydrolase 2	1.57	2.60	0.32	0.29	0.61	0.047	−0.72	1.33
Q5T6V5	C9orf64	UPF0553 protein C9orf64	1.11	1.80	0.19	0.24	0.62	0.047	−0.70	1.33
P01034	CST3	Cystatin-C	1.55	2.51	0.27	0.31	0.62	0.041	−0.69	1.39
Q7Z4V5	HDGFRP2	Hepatoma-derived growth factor-related protein 2	1.11	1.78	0.20	0.21	0.62	0.044	−0.69	1.36
B3KM48	-	cDNA, FLJ10286fis, clone HEMBB1001384, highly similar to COP9 signalosome complex subunit 4	1.52	2.41	0.25	0.32	0.63	0.049	−0.67	1.31
P62312	LSM6	U6 snRNA-associated Sm-like protein LSm6	1.24	1.89	0.22	0.13	0.65	0.043	−0.61	1.37
A0A087WYV5	SLIT2	Slit homolog 2 protein	1.25	1.88	0.06	0.09	0.67	<0.0002	−0.58	3.90
P08294	SOD3	Extracellular superoxide dismutase (Cu-Zn)	0.87	1.3	0.08	0.18	0.67	0.035	−0.58	1.46
***A0A024R884***	***TNC***	***Tenascin C (Hexabrachion), isoform CRA_a***	***1.88***	***0.68***	***0.29***	***0.15***	***2.75***	***0.010***	***1.46***	***2.02***
***A0A0C4DFX3***	***EMILIN1***	elastin microfibril interfacer 1	***2.90***	***1.64***	***0.39***	***0.27***	***1.76***	***0.038***	***0.82***	***1.42***

Proteins that are found at higher levels in BAV are shown in bold italics at the bottom of the table. #; number.

**Table 2 jcm-08-00517-t002:** Significant canonical pathways with enriched protein expression.

IPA Canonical Pathway	*p*-Value	Molecule
Superoxide radicals degradation	1.16 × 10^−2^	SOD3
D-myo-inositol (1,4,5) Trisphosphate Biosynthesis	3.85 × 10^−2^	PLCD1

**Table 3 jcm-08-00517-t003:** Diseases and functions with enriched protein expression for significantly altered total protein levels in aortic tissue of BAV patients compared to TAV patients.

IPA Disease or Function	*p*-Value	Molecules	# Molecules
**Overall**			
Morphology of hippocampal neurons	9.13 × 10^−5^	CST3, SYNCRIP	2
Formation of elastin fibres	1.12 × 10^−4^	EMILIN1, TNXB	2
Secondary Tumour	1.41 × 10^−4^	ADIRF, CST3, MAP4, PSMD10, PTPRD, SLIT2, SNCG, SOD3, TNC	9
Migration of oligodendrocyte precursor cells	1.58 × 10^−4^	SLIT2, TNC	2
Morphogenesis of aortic valve	3.44 × 10^−4^	EMILIN1, SLIT2	2
**Cardiovascular System Development and Function**			
Morphogenesis of aortic valve	3.44 × 10^−4^	EMILIN1, SLIT2	2
Relaxation of carotid artery	1.01 × 10^−2^	SOD3	1
Angiogenesis	1.58 × 10^−2^	CST3, EMILIN1, ERAP1, PLCD1, SLIT2, TNC	6
Relaxation of aorta	2.02 × 10^−2^	SOD3	1
Abnormal morphology of tunica media	2.16 × 10^−2^	CST3	1
Vasculogenesis	2.43 × 10^−2^	CST3, ERAP1, PLCD1, SLIT2, TNC	5
Vascularization of placenta	2.44 × 10^−2^	PLCD1	1
Abnormal morphology of aortic arch	4.27 × 10^−2^	CST3	1

The top half of the table shows the top five most significant diseases and functions overall. The bottom half of the table shows all the significant diseases and functions under the IPA category ‘Cardiovascular System Development and Function’. #; number.

**Table 4 jcm-08-00517-t004:** Significantly changed phosphoproteins in the aortas of neonatal BAV patients compared to TAV patients.

Accession #	Gene ID	Protein Name	Protein Phosphosite	Mean		SEM		Fold Change (BAV/TAV)	*p*-Value	log_2_ (Fold Change)	−log_10_ (*p*-Value)
				BAV	TAV	BAV	TAV				
C9JUG3	SSH3	Protein phosphatase slingshot homolog 3	Serine 3	1.11	1.82	0.18	0.2	0.61	2.4 × 10^−2^	−0.72	1.619

Protein phosphosite: the number indicates the position of the modification in the protein.

**Table 5 jcm-08-00517-t005:** Canonical pathway analysis: pathways likely to be altered by SSH3 phosphorylation.

IPA Canonical Pathway	*p*-Value
D-myo-inositol (1,4,5,6)-tetrakisphosphate Biosynthesis	6.54 × 10^−3^
D-myo-inositol (3,4,5,6)-tetrakisphosphate Biosynthesis	6.54 × 10^−3^
3-phosphoinositide Degradation	7.17 × 10^−3^
D-myo-inositol-5-phosphate Metabolism	7.35 × 10^−3^
3-phosphoinositide Biosynthesis	9.17 × 10^−3^
Actin Cytoskeleton Signalling	1.02 × 10^−2^
Superpathway of Inositol Phosphate Compounds	1.08 × 10^−2^

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
