# Peer review of "Bicuspid Aortic Valve Alters Aortic Protein Expression Profile in Neonatal Coarctation Patients"

_jcm, 2019, doi:10.3390/jcm8040517_

Round 1
Reviewer 1 Report
This is a well-written manuscript describing a proteomic analysis of coarcted aorta in neonatal patients with a normal tricuspid aortic valve or a congenitally abnormal bicuspid aortic valve. Only minor revisions are requested:
- Line 55: replace endocardial with endothelial.
- Line 252: provide a citation to support the statement.
- Line 263: replace SOD expression by activity. Citation #25 did not measure SOD expression.
Author Response
Responses for Reviewer 1
This is a well-written manuscript describing a proteomic analysis of coarcted aorta in neonatal patients with a normal tricuspid aortic valve or a congenitally abnormal bicuspid aortic valve. Only minor revisions are requested:
Line 55: replace endocardial with endothelial.
Apologies for this mistake, we have made the change as suggested.
Line 252: provide a citation to support the statement.
We have now added in a reference to support this statement.
Line 263: replace SOD expression by activity. Citation #25 did not measure SOD expression.
Thank you, we have made the change as suggested.
Reviewer 2 Report
Among the remarks and commentaries:
This manuscript focuses on a small “double” cohort of patients presenting with the neonatal form of aortic coarctation.
It is well written. The topic is modern, and obviously of interest.
As very few data have been reported on the proteonomics in human neonatal aortic tissue, this one is per se interesting to read.
A weakness of the study is the fact that proteonomics where performed in 12 patients, rather evenly distributed between tricuspid and bicuspid valve phenotype, whereas histologic and morphometry was done in another group of 11 patients.
Ideally, all individuals should have undergone both analysis, so that associations would have been stronger.
A control group of aortic wall from neonates without coarctation undergoing ASO could have been added to discriminate the role of isthmic coarcation on aortic wall proteonomics.
There is no dichotomization of the results whether the sample was taken from the isthmus, or the distal aortic arch, though the presence of myofibroblasts and other ductal cells could have modified the results. Intra- and inter-group variablility based on the location of the sample could have brought additional information.
In their discussion, the authors states in the BAV phenotype is responsible for altered hemodynamics, which in turn is likely to explain the differences observed in the gene and protein expression.
Before birth, the aortic blood flow represent only 40% of the entire cardiac output with 60% of the flow going directly through the ductus arteriosus. It is unlikely that the BAV phenotype can influence at this early stage the proteonomics.
At birth, most of those neonates will receive PGE1 infusion prior to surgery so that again, the flow through the aortic valve is less than theoretical, and Doppler-flow measurements are almost invariably normal, with Vmax less 2m/sec.
Correlation and causative relationship are very distinct, and the authors should probably be more cautious in their comments and hypothesis.
Author Response
We would like to thank the reviewer for his/her constructive feedback and comments which were extremely helpful. We have now revised the manuscript, taking into account all the comments made by the reviewer.
Among the remarks and commentaries:
This manuscript focuses on a small “double” cohort of patients presenting with the neonatal form of aortic coarctation. We agree it is a small cohort. However, recruiting neonates and collecting tissue from them is a very sensitive issue and takes much longer than older patients.
It is well written. The topic is modern, and obviously of interest. As very few data have been reported on the proteonomics in human neonatal aortic tissue, this one is per se interesting to read. We would to thank the reviewer for this comment.
A weakness of the study is the fact that proteonomics where performed in 12 patients, rather evenly distributed between tricuspid and bicuspid valve phenotype, whereas histologic and morphometry was done in another group of 11 patients. Ideally, all individuals should have undergone both analysis, so that associations would have been stronger. As mentioned above, there are severe restrictions involved in collecting tissue from neonates. The size of the tissue samples which can be obtained is also very small and would be difficult to use for both measurements. Additionally, there could be heterogeneous changes in different part. It is possible however to do this in infants or children where larger pieces of tissues could be collected. We have amended the manuscript to include this in our limitations section (lines 277-285).
A control group of aortic wall from neonates without coarctation undergoing ASO could have been added to discriminate the role of isthmic coarcation on aortic wall proteonomics. Strictly speaking, aortic tissue collected from patients undergoing an arterial switch operation is likely to have molecular remodelling in response to its own congenital disease. We have previously demonstrated that disease in one chamber can trigger molecular remodelling in other chambers of the heart (J Proteomics. 2019 Jan 16;191:107-113), so we wouldn’t be able to tell if observed differences between the groups were due to the lack of a coarctation, or the presence of other abnormalities. As we have now stated in our limitations section (lines 277-285), the best control group would of course be tissue from healthy individuals, which unfortunately is not possible.
There is no dichotomization of the results whether the sample was taken from the isthmus, or the distal aortic arch, though the presence of myofibroblasts and other ductal cells could have modified the results. Intra- and inter-group variablility based on the location of the sample could have brought additional information. In this study, all tissues were collected from above the coarctation area, so it is pre ductal tissue. We have amended the methods to make this slightly clearer (lines71-72). We agree it is possible that slight regional differences in the tissue collection could have introduced variation in the results, and we have now mentioned this in our limitations section. Comparing different regions of the aorta could be a target of a future study.
In their discussion, the authors states in the BAV phenotype is responsible for altered hemodynamics, which in turn is likely to explain the differences observed in the gene and protein expression. Before birth, the aortic blood flow represent only 40% of the entire cardiac output with 60% of the flow going directly through the ductus arteriosus. It is unlikely that the BAV phenotype can influence at this early stage the proteonomics. At birth, most of those neonates will receive PGE1 infusion prior to surgery so that again, the flow through the aortic valve is less than theoretical, and Doppler-flow measurements are almost invariably normal, with Vmax less 2m/sec. We agree that the velocity of blood flow through the aortic valve will be low. However it is not only the velocity that is important; studies have suggested that the coexistence of CoA and BAV will cause increases in aortic turbulent flow and shear stress (e.g. PLoS One. 2013 Aug 27;8(8):e72394). Changes in proteomics can occur quite quickly, and certainly within three weeks, therefore it is possible that such changes in blood flow haemodynamics could affect the neonatal proteome. We have amended the discussion (lines 216-221) and the introduction (lines 46-48, 63-64) to make this point clearer.
Correlation and causative relationship are very distinct, and the authors should probably be more cautious in their comments and hypothesis. Thank you for this comment. We have now altered the discussion and introduction to remove implications of causation which we do not have evidence for (lines 252-254, 266-268, 63-64).
Reviewer 3 Report
Is a very interesting topic. The paper is well write. Good methodology. I suggest the publication with only 1 minor revision.
1) Add limitation study section in discussion
Author Response
Responses for Reviewer 2
Is a very interesting topic. The paper is well write. Good methodology. I suggest the publication with only 1 minor revision.
1) Add limitation study section in discussion
Thank you for your suggestion, we have added in a limitation section in the discussion (line 276).